# Structural dynamics of SARS-CoV-2 nucleocapsid protein induced by RNA binding

**Helder Veras Ribeiro-Filho**[1☯], **Gabriel Ernesto Jara**[1☯], **Fernanda Aparecida Heleno Batista**[1], **Gabriel Ravanhani Schleder**[2], **Celisa Caldana Costa Tonoli**[1], **Adriana Santos Soprano**[1], **Samuel Leite Guimarães**[1], **Antonio Carlos Borges**[2], **Alexandre Cassago**[2], **Marcio Chaim Bajgelman**[1], **Rafael Elias Marques**[1], **Daniela Barretto Barbosa Trivella**[1], **Kleber Gomes Franchini**[1], **Ana Carolina Migliorini Figueira**[1], **Celso Eduardo Benedetti**[1]\*, **Paulo Sergio Lopes-de-Oliveira**[1]\*

**1** Brazilian Biosciences National Laboratory, Brazilian Center for Research in Energy and Materials (CNPEM), Campinas, Brazil, **2** Brazilian Nanotechnology National Laboratory, Brazilian Center for Research in Energy and Materials (CNPEM), Campinas, Brazil

☯ These authors contributed equally to this work.
\* celso.benedetti@lnbio.cnpem.br (CEB); paulo.oliveira@lnbio.cnpem.br (PSLO)

**Data Availability Statement:** All data related to the computational simulations are available from Zenodo repository (DOI: 10.5281/zenodo. 5841603). All other data needed to evaluate the

## Abstract

The nucleocapsid (N) protein of the SARS-CoV-2 virus, the causal agent of COVID-19, is a multifunction phosphoprotein that plays critical roles in the virus life cycle, including transcription and packaging of the viral RNA. To play such diverse roles, the N protein has two globular RNA-binding modules, the N- (NTD) and C-terminal (CTD) domains, which are connected by an intrinsically disordered region. Despite the wealth of structural data available for the isolated NTD and CTD, how these domains are arranged in the full-length protein and how the oligomerization of N influences its RNA-binding activity remains largely unclear. Herein, using experimental data from electron microscopy and biochemical/biophysical techniques combined with molecular modeling and molecular dynamics simulations, we show that, in the absence of RNA, the N protein formed structurally dynamic dimers, with the NTD and CTD arranged in extended conformations. However, in the presence of RNA, the N protein assumed a more compact conformation where the NTD and CTD are packed together. We also provided an octameric model for the full-length N bound to RNA that is consistent with electron microscopy images of the N protein in the presence of RNA. Together, our results shed new light on the dynamics and higher-order oligomeric structure of this versatile protein.

## Author summary

The nucleocapsid (N) protein of the SARS-CoV-2 virus plays an essential role in virus particle assembly as it specifically binds to and wraps the virus genomic RNA into a well-organized structure known as the ribonucleoprotein. Understanding how the N protein wraps around the virus RNA is critical for the development of strategies to inhibit virus assembly within host cells. One of the limitations regarding the molecular structure of the ribonucleoprotein, however, is that the N protein has several unstructured and mobile

conclusions in the paper are present in the main manuscript and the Supplementary Materials.

**Funding:** This work is part of the Rede Virus MCTI taskforce on COVID-19 funded by FINEP (grant number 01.20.0003.00) (http://www.finep.gov.br/), Brazilian Ministry of Science, Technology and Innovation (https://www.gov.br/mcti). GRS received financial support from the Fundação de Amparo à Pesquisa do Estado de São Paulo (FAPESP - https://fapesp.br/), project number 17/18139-6. The funders had no role in study design, data collection and analysis, decision to publish, or preparation of the manuscript.

**Competing interests:** The authors have declared that no competing interests exist.

regions that preclude the resolution of its full atomic structure. Moreover, the N protein can form higher-order oligomers, both in the presence and absence of RNA. Here we employed computational methods, supported by experimental data, to simulate the N protein structural dynamics in the absence and presence of RNA. Our data suggest that the N protein forms structurally dynamic dimers in the absence of RNA, with its structured N- and C-terminal domains oriented in extended conformations. In the presence of RNA, however, the N protein assumes a more compact conformation. Our model for the oligomeric structure of the N protein bound to RNA helps to understand how N protein dimers interact to each other to form the ribonucleoprotein.

## Introduction

All coronaviruses, including the Severe Acute Respiratory Syndrome Coronavirus 2 (SARS--CoV-2), the causative agent of the Coronavirus disease 2019 (COVID-19) pandemics, possess an organized nucleocapsid formed by a ribonucleoprotein (RNP) complex surrounded by a lipid envelope [1–3]. The major component of the RNP complex is the nucleocapsid (N) protein, one of the four structural proteins of coronaviruses and also the most abundantly expressed viral protein in infected host cells [2].

N proteins are conserved among coronaviruses and are known to play multiple roles in the virus life cycle [4]. In addition to packaging the viral genomic RNA, N proteins are required for genome replication, transcription and translation, and for the assembly of the RNPs into newly formed viral particles [5–13]. This functional diversity is intimately linked to the dynamic structure of the N protein and its ability to bind and alter the RNA structure [9,14].

Coronaviruses N proteins are composed of two structured and globular domains represented by the N- (NTD) and C-terminal (CTD) domains, both of which are capable of binding single-stranded RNA and DNA molecules [15–24]. The NTD has an extensive basic U-shaped RNA-binding cleft implicated in the binding and melting of transcription regulatory sequences (TRS) needed for transcription of sub-genomic RNAs [9,14]. The CTD is responsible for the protein dimerization and it also forms a positively charged groove thought to contribute to the recognition of the packaging signal (PS) and to the assembly of the RNP into the virion particle [13,17,23]. In addition to the CTD, the flexible C-terminal tail also seems to influence protein oligomerization by promoting protein tetramerization [25–27]

The NTD and CTD are connected by a central disordered serine and arginine-rich region, denoted as SR linker. This linker region is also proposed to play fundamental roles in protein oligomerization and function. Of note, the SR linker was shown to be modified by phosphorylation, which not only reduces the affinity of the protein for the RNA, but also drives a liquid phase separation of the N protein with the RNA and other virus proteins and host cell components [28–30]. Recently, SR-linker mutations have been associated with the emergence of the high-transmissible SARS-CoV-2 lineage B.1.1.7 [31]. Therefore, understanding how the NTD and CTD are oriented in structure and how the SR-linker and other unstructured regions contribute to the overall N protein organization is of paramount importance.

Due to its dynamic structure and multiple oligomeric organization depending on environmental conditions, and to the fact that the N protein is also modified by phosphorylation [28,29,32], no three-dimensional (3D) structures are available for coronaviruses full-length N proteins. Here, by combining electron microscopy and biophysical experimental analysis with molecular modeling and molecular dynamics simulations, we propose structural models for the full-length SARS-CoV-2 N protein in the absence and presence of RNA. These models not

only support current experimental data, but also provide a framework for understanding the multifunctional role of the N protein.

## Results

### Dimers of full-length N adopts an extended conformation in the absence of RNA

To shed light on the structure of full-length N and to understand how the NTD and CTD are oriented in structure, the recombinant N protein produced in *E. coli* was purified by affinity and size-exclusion chromatography (SEC) and analyzed by negative stain electron microscopy (NSEM). We noticed that the N protein, which appeared as a single peak in SEC and as a single band in denaturing gel electrophoresis (Fig 1A), contained traces of bacterial RNA in non-denaturing gels, since treatments with RNase A, but not Dnase I, removed most of the nucleic

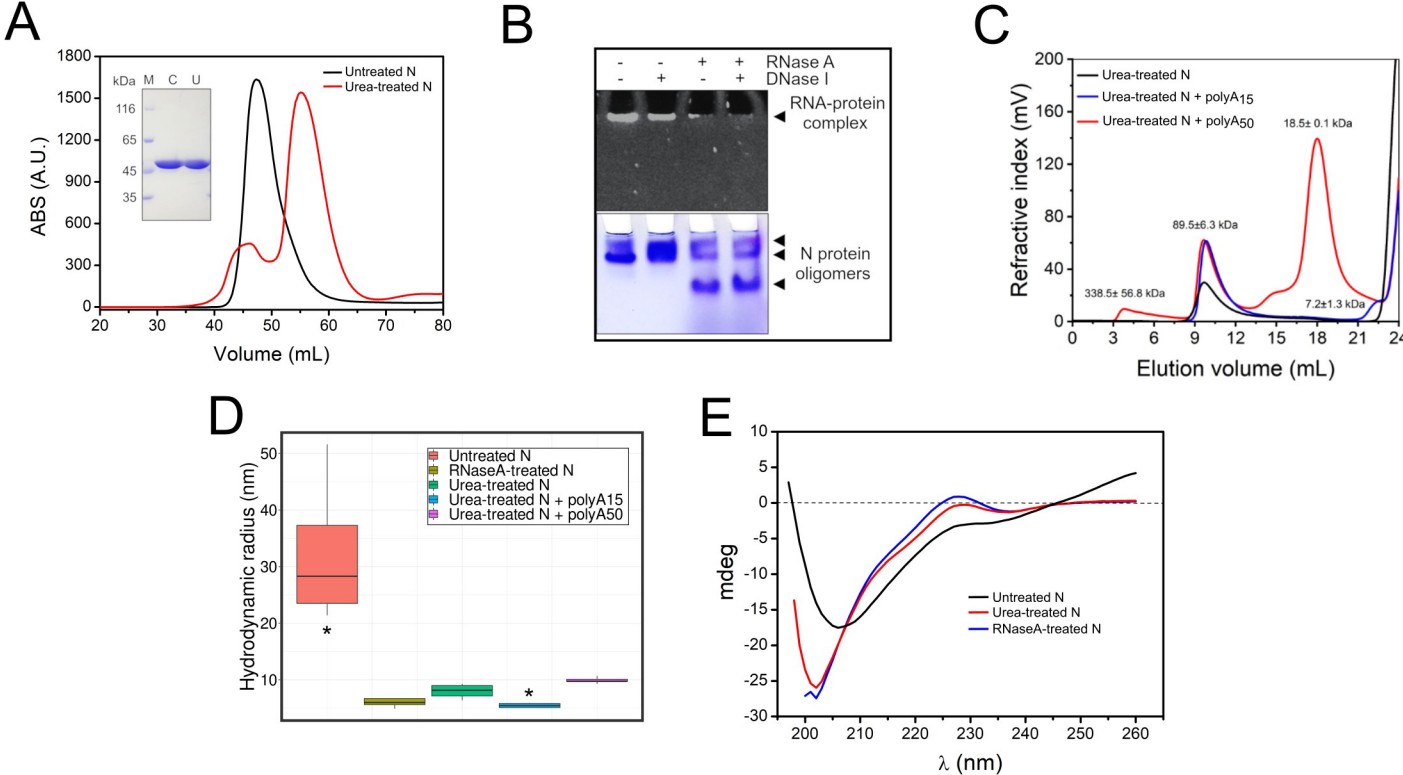

**Fig 1. The N protein forms high molecular-weight complexes with RNA but is a dimer in solution in the absence of RNA.** A- SEC showing that the untreated N protein sample (black line) elutes as a single peak in the void volume of the gel filtration column whereas the urea-treated sample (red line) elutes as a major peak of a higher elution volume. The purity of these proteins samples was evaluated by SDS-PAGE, which shows single bands of the expected size for the recombinant N protein (inset). For urea-treated samples, all experimental analysis were carried out using fractions corresponding to the center of the major peak, since the minor peak likely corresponds to the protein still carrying nucleic acid. B- Native gel electrophoresis stained with ethidium bromide (upper panel) or Coomassie blue (lower panel) showing that the N protein forms high molecular-weight complexes with bacterial RNA, as treatments with RNase A, but not DNase I, dissolve these complexes (arrowhead). The RNase A treatment also changed the oligomeric forms of N protein (arrowheads) in solution (lower panel). C- SEC-MALS analysis of urea-treated N protein in the absence or presence of polyA15 or polyA50. The chromatogram shows that the N protein treated with urea as well as in the presence of polyA15 are dimers in solution, with a mean molecular mass of 89.5 ± 6.3 kDa. In addition to the peak of dimers, urea-treated N presents a higher molecular-weight peak (338.5 ± 56.8 kDa) and a lower molecular-weight peak (18.5 ± 0.1 kDa) D- DLS measurements showing the Rh of untreated N protein carrying bacterial RNA (red boxplot), N protein treated with RNase A (brown boxplot) and N protein treated with urea in the absence (green box plot) or presence of polyA15 (blue boxplot) or polyA50 RNA (purple boxplot). Statistical comparison was performed using Dunnett-Tukey-Kramer pairwise multiple comparison test adjusted for unequal variances and unequal sample sizes and P value was estimated from the confidence interval (see Dynamic Light Scattering in Methods for details). *P<0.05 comparing urea-treated N with the other conditions. E- CD plot showing that CD curves of N protein samples treated with RNase A (blue) and urea (red) are compatible These curves are more similar to each other than the one of N protein samples carrying bacterial RNA (black).

acid associates with the protein (Fig 1B) and reduced the 260/280 nm absorption ratio from 1.8 to 0.6. Most notably, removal of the contaminating RNA with RNase A led to a change in the oligomeric state of the protein (Fig 1B).

To investigate how the N protein behaves in the absence of RNA, the recombinant protein was treated with urea and high salt concentration to remove bound RNA [28,33,34], and purified by affinity chromatography and SEC. The urea-treated N, which presented a 260/280 nm absorption ratio of ~0.5, eluted as a major peak with a higher elution volume compared with the untreated protein (Fig 1A). This sample showed a molecular mass of 89,5 ± 6.3 kDa and hydrodynamic radius (Rh) of 8.0 ± 1.3 nm determined by SEC-MALS and DLS, respectively, consistent with a dimer in solution (Fig 1C and 1D). Noteworthy, the Rh is close to the one recently reported for the N protein purified under similar conditions [35]. Also, these results are in line with literature data showing that N protein readily oligomerizes into dimers [36,37]. The urea-treated N showed a circular dichroism profile comparable to that of the protein treated with RNase A (Fig 1E), indicating that the urea treatment did not substantially alter the protein secondary structure. Both curves are similar to previously reported N protein CD profiles [37]. On the other hand, the CD profile of untreated N protein presents a positive ellipticity at 250–260 nm which represents the contribution of RNA in this sample.

Negative stain images produced from the urea-treated N samples showed various rounded particles with dimensions smaller than 5 nm (Fig 2A). Thus, the dimensions of these particles are not compatible with the Rh determined experimentally for the full-length N protein (Fig 1D). Nevertheless, these 5 nm-particles can represent the globular domains of the N protein in an extended conformation, in which each particle corresponds to individual NTD or CTD units. The N protein disordered regions, including the SR-linker and the N and C-terminal tails, do not present sufficient contrast in the NSEM images to be distinguished from the background noise.

To investigate this hypothesis, we picked 3500 particles and performed a two-dimensional (2D) classification into 100 classes to calculate the area of each class average (Fig 2A, inset). We compared the distribution of areas of the urea-treated N class averages with the distribution of areas of simulated reprojections derived from the monomeric NTD and dimeric CTD atomic structures (Fig 2B) (see Tools for comparing CG simulations with biophysical or electron microscopy data in Methods and S2B Fig for details). Notably, we found that the area distributions of the urea-treated N protein fitted well into the area distributions of the NTD monomer and CTD dimer (Fig 2B). This suggested that the particles observed in the NSEM images (Fig 2A) likely correspond to isolated NTD and CTD globular domains of the N protein dimer in extended conformations. The idea that N protein adopts an extended conformation in absence of RNA is supported by coarse-grained (CG) molecular dynamics simulations of the full-length N protein dimer, which show that the NTD regions move freely in relation to the CTD dimer, producing a variety of conformers (Fig 2C) with an estimated mean radius of gyration (Rg) of 6.6 nm (Fig 2D). This value is close to the Rg of a truncated SARS-CoV-1 N protein without the N- and C-terminal tails (6.1nm), determined by SAXS [19]. The Rg calculated from CG simulations is numerically different from the experimentally estimated Rh described above, which is expected since they represent two different physical properties [38,39]. Thus, in our CG simulations, we used the Rg values as a parameter to estimate the N protein compaction upon RNA binding, while Rh values were used for comparing protein compaction in DLS experiments.

The CG simulations do not point to NTD-CTD contacts (S1 Fig), supporting the idea that the large density of positive charges on the surface of the NTD and CTD causes repulsion of these domains, which is favored by the flexibility of the SR linker. The simulated 2D reprojections derived from representative 3D models (with minimum, mean or maximum Rg) of the

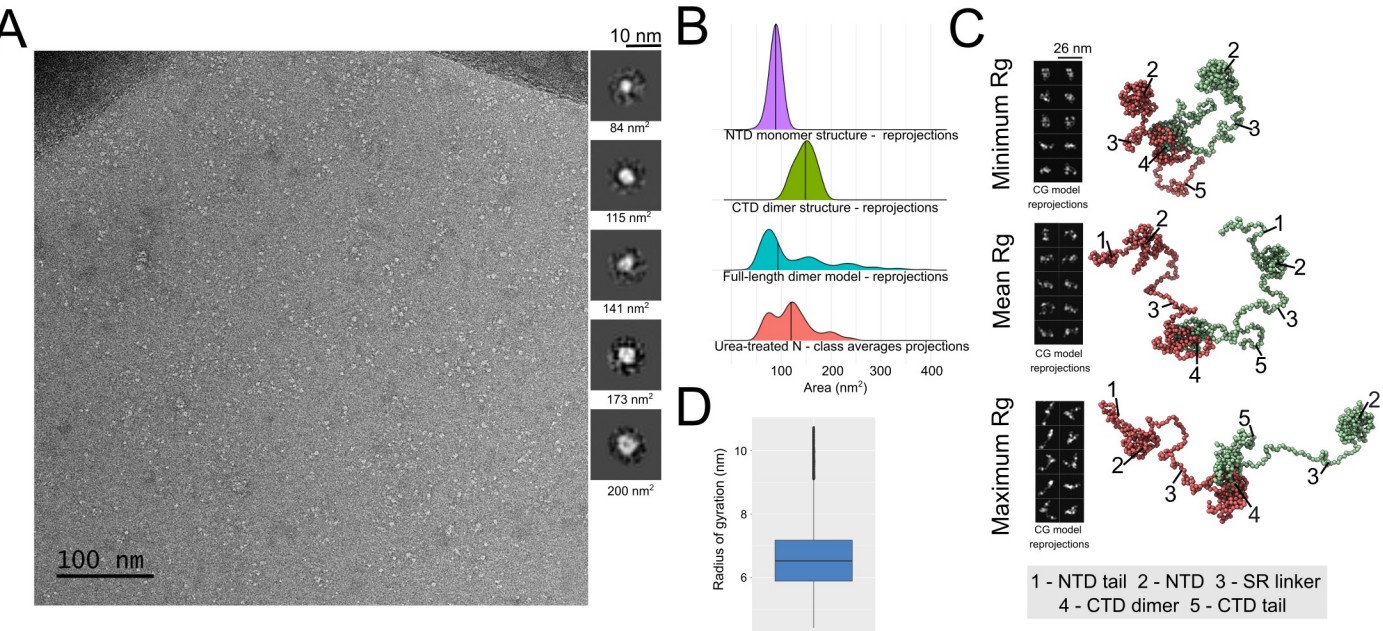

**Fig 2. Dimers of full-length N show an extended conformation in the absence of RNA.** A- Representative negative stain image of urea-treated N in the absence of RNA. Insets on the right present representative class averages from a total of 100 classes produced by the 2D classification of 3500 particles. The area corresponding to each particle is presented. B- Density plots of area distribution estimated for simulated 2D reprojections from NTD monomer structure (PDB ID: 6M3M), CTD dimer structure (PDB ID: 6WZO) or full-length dimer reprojections, or for class averages of urea-treated N. In the case of full-length dimer reprojections, when N protein structured domains (NTD or CTD) were presented as separated density regions in the reprojections, the area of these regions were computed separately. For this reason, we can see three main peaks in full-length N dimer distribution: the first peak corresponding to isolated NTD domains, the second to isolated CTD domains and the third to combinations of these domains close to each one in the 2D reprojected image (see Tools for comparing CG simulations with biophysical or electron microscopy data in Methods for details). C- Representative frames of the minimum (upper panel), mean (middle panel) or maximum (lower panel) Rg obtained from the CG simulations of the full-length N protein dimer. 2D simulated reprojections from each frame that were used to estimate area are presented at the left. N protein domains are numbered. D- Boxplot of the Rg estimated from five independent CG simulations of full-length N protein dimer in absence of RNA.

CG simulation also illustrate the diversity of N protein conformations. These simulated 2D reprojections show density regions of similar size of the particles observed in the NSEM images (Fig 2A and 2C). In the case of full-length N dimer conformations obtained from CG, depending on its orientation, the reprojections of these conformations can be seen as three high density points that correspond to the structured domains (two NTD and one CTD dimer) in an extended conformation (Figs 2C and S2B), producing the multimodal distribution observed in Fig 2B. The area distributions of the urea-treated N class averages overlap with the area distributions of the simulated reprojections of the full-length N dimer (Fig 2B). Together, our data suggest that dimers of full-length N show extended and highly flexible conformations in the absence of RNA.

## Distinct forms of the N protein bound to RNA were revealed by NSEM

Because no 3D structures of full-length N in complex with RNA are available and only few microscopy studies have attempted to report the structural organization of this N protein [28], we inspected the N protein samples not treated with urea or RNase A by NSEM. The NSEM micrographs revealed that the N protein in samples carrying primarily bacterial RNA (Fig 1B) contained a myriad of particles and aggregates that are consistent in size with the DLS measurements (Figs 1D and 3).

The larger aggregates with a width of 20–30 nm (Fig 3A) resemble the helical structures of the linearly arranged N protein isolated from the transmissible gastroenteritis (TGEV)

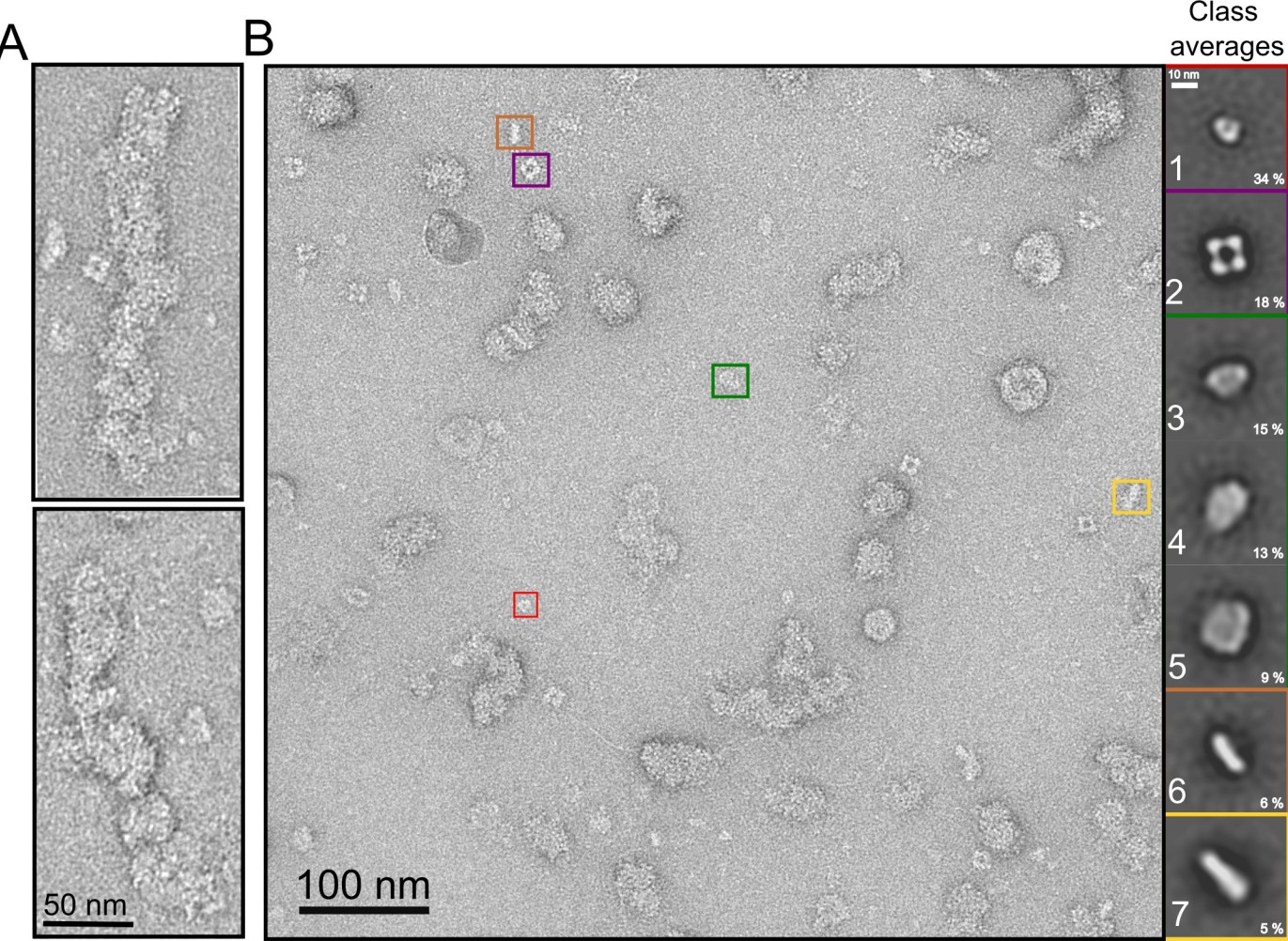

**Fig 3. Representative negative stain micrographs of untreated N protein samples containing bacterial RNA.** A- Detail of large helical-like structures with 20–30 nm width. B- Representative micrograph showing a wide range of particles and aggregates. Smaller particles, with less than 20 nm in diameter are shown (colored boxes). In insets, class averages (1 to 7) of picked particles with sizes up to 20 nm in width from one hundred electron micrographs: class-1 (toroid-like particles), class-2 (square-like particles), classes 3 to 5 (elliptical and round-like particles), and classes 6 and 7 (rod-like particles). The percentage of each class average in relation to all picked particles are indicated.

coronavirus [40]. Smaller particles with less than 20 nm in diameter were also observed (Fig 3B, colored boxes). Because these particles were quite abundant and structurally diverse, and showed dimensions consistent with N protein particles isolated from SARS-CoV-2 and related virus [41–43], we classified them according to their size and shape (Fig 3B, inset).

The 2D class averages analysis revealed seven particle classes (Fig 3B). Class-1 (toroid-like) particles comprised rounded particles with a weak central density. These particles, which were the smallest and most compact (between 7 and 10 nm wide), resemble those corresponding to the mouse hepatitis virus (MHV) N protein dimer [41,42].

Class-2 particles comprised square-like particles approximately 14 nm wide. Notably, these particles have not yet been associated with the SARS-CoV-2 N protein; however, they resemble the RNPs isolated from the human coronavirus 229E (HCV229E) and MHV3[44]. The class-2 particles also resemble those of the M1 influenza A protein and Schmallenberg orthobunyavirus RNP, where each globular unit located at the square vertices corresponded to a single structured protein domain [45,46].

Class averages 3 to 5 comprised elliptical and rounded particles ranging from 11 to 19 nm in length, whereas class averages 6 and 7 comprised rod-like structures of 16–20 nm in length (Fig 3B).

Although the diversity of these particles hinders the use of classical single-particle approaches needed for 3D structure high-resolution, we used the 2D class average analysis to estimate particle dimensions and to guide molecular modeling and dynamics simulations to gain insights into the conformational states of full-length N bound to RNA. We focused our analysis on the most abundant particles represented by classes 1 and 2, which are also the particles that showed well-defined features and dimensions (Fig 3B).

## N protein dimers adopt a packed conformation in the presence of RNA

As reported recently [36,37] and shown in Fig 1C, the N protein devoid of RNA is a dimer in solution, which is consistent with the notion that dimers are the fundamental oligomeric unit of full-length N. The toroid-like particles observed in Fig 3B had dimensions ranging from 7 to 10 nm, which were smaller than the expected N protein diameter estimated from Rh (2 x 8.0 nm in diameter) or Rg (2 x 6.6 nm in diameter) as described above. Thus, we reasoned that these particles could represent N protein dimers in a more compact conformation due to the presence of RNA. This idea is supported by the observation that the RNase A-treated N, which may still have RNA bound to it, exhibited a smaller Rh (5,9 ± 0,7 nm) compared with the urea-treated protein. (Fig 1D).

To investigate this hypothesis, the urea-treated N was incubated with a polyA15 or polyA50 RNA and the Rh was determined by DLS. We found that the N protein incubated with polyA15, but not polyA50, showed a significant reduction in the Rh (5.5 ± 0.4 nm), compared with the RNA-free N protein (Fig 1D). Notably, the Rh observed for the protein in the presence of polyA15 was comparable to that of the RNase A-treated protein, which still retains RNA (Fig 1D). On the other hand, the presence of polyA50 RNA increased the Rh of N protein in comparison to the presence of polyA15, producing a Rh slightly higher than the RNA-free N protein (Fig 1D). Interestingly, SEC-MALS analysis of the urea-treated N protein incubated with the polyA50 showed, in addition to the N dimer peak, a small but detectable peak of approximately 340 kDa, which would be consistent with an N protein octamer (Fig 1C). Such peak was not observed with polyA15. In addition, when inspected by NSEM, N protein samples incubated with polyA50 showed larger particles of irregular shape in comparison to samples with polyA15 (S2A Fig), which would infer a long RNA could join two or more dimers of N. These results are in line with the increase in Rh produced by polyA50 and thus suggest that the N protein undergoes protein compaction and possibly oligomerization in the presence of RNA.

To understand how the RNA could drive protein compaction, we performed CG dynamics simulations of dimeric full-length N, in which protein-RNA contacts are mostly driven by electrostatic forces. The simulations were carried out in the presence of single-strand RNA molecules ranging from 10 to 70 nucleotides (nt) in length (Fig 4). We observed that while the CTD dimer, which has the highest density of positive charges, readily associates with the shortest RNA molecules (10–20 nt), the NTD units move freely, with no contact with the RNA (Fig 4A and 4C). According to our simulations, a single short RNA molecule is unable to interact simultaneously with the NTD and CTD. Furthermore, since the NTD units moved freely relative to CTD, the Rg distribution of the protein in complex with the shortest RNA is similar to that of the protein without RNA (Fig 4B). Two main CTD regions interact with the RNA molecule in these simulations: The first region (240–280) has a high density of positively charged residues, while the second region, although with less positively charged residues, is structurally

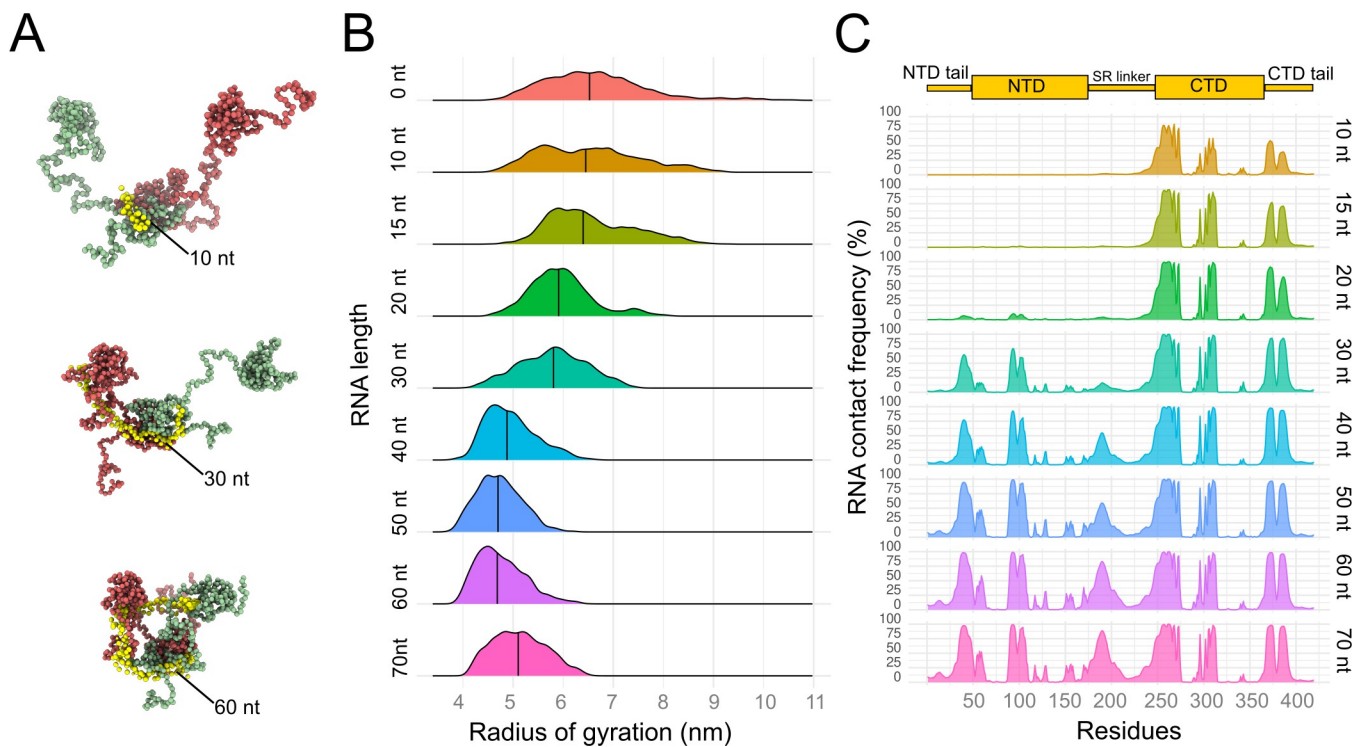

**Fig 4. CG dynamics simulations of full-length N dimers in the presence of RNA of varying lengths.** A- Representative frames, obtained from CG simulation, of mean Rg for N protein dimer in complex with a single strand RNA molecule with 10, 30 or 60 nucleotides. N protein monomers are colored in red and green whereas the RNA is colored in yellow. B- Density plot of the Rg distribution (in nm) calculated from five independent CG simulations of the N protein dimers in the presence of single-strand RNA of 10 to 70 nucleotides (nt) long. The median of distribution is indicated by a vertical line. C–Density plot of the frequency of contacts between N protein and any RNA nucleotide during the CG simulation. For instance, a frequency of 100% for N protein residue 260 means that this residue, from any of the N protein monomers, makes contact with at least one nucleotide of the RNA segment in all the analyzed frames of the CG simulations. The distance cut-off used to consider a protein-RNA contact in CG model was 10 Å between the alpha carbon of the protein residues and any atom of the RNA.

close to the first region. The simulations also point to interactions of the CTD tail (370–390) with the RNA. It was found that RNAs with 30 or more nucleotides are long enough to simultaneously bind the CTD dimer and at least one unit of the NTD (Fig 4A and 4C). The RNA-binding regions involving the NTD include residues 30 to 50 and 90 to 110. Additional contact regions including the SR-linker (180–200) are also seen with RNAs longer than 40 nt (Fig 4C). This seems to be the cause of the shift to a smaller Rg observed in the distribution (Fig 4A). As the length of the RNA molecules increases (50 to 60-nt), a single RNA molecule can interact simultaneously with the two NTD units and one CTD dimer bringing these domains together. Full compaction of the N protein dimer was observed with the 40 to 60-nt RNA models with a minimum Rg of 3.7 nm for these models (Fig 4B). However, the presence of multiple RNA binding sites along the N protein could account for the compaction observed experimentally even with shorter RNA such as the polyA15 RNA (Fig 1D). Indeed, CG simulations of the N protein dimer with multiple RNA molecules of 15 nt also presented a reduction in the protein Rg relative to the RNA-free protein (S3 Fig), which is thus in line with the DLS results (Fig 1D).

To better correlate the CG simulations with the negative stain images, we built a low-resolution 3D density map from the toroid-like particles (Fig 5). The comparison between the class averages used to build the 3D map and reprojections generated from the map is presented in S4A Fig. Next, we leveraged the conformational sampling obtained from the CG simulations

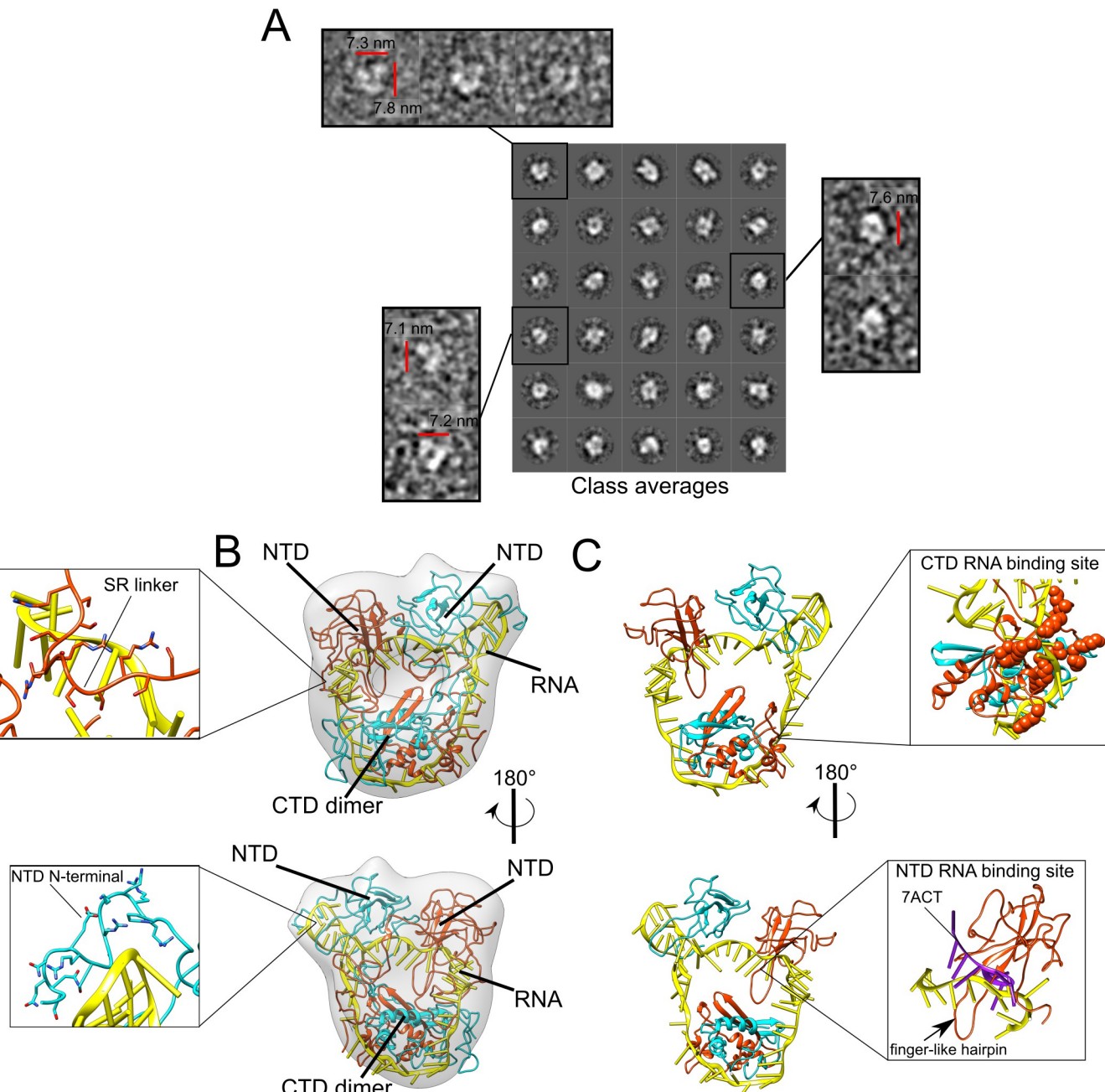

**Fig 5. The full-length N dimer undergoes domain compaction in the presence of RNA.** A- 2D classification of 24137 toroid-like particles picked with xmipp3. Thirty class averages used for the 3D model are shown in the main panel, whereas raw projections of particles representing the classes are presented in the insets. B- 3D density map reconstruction of the toroid-like particles with the flexible fitting of the N dimer atomic model derived from the CG simulations performed with the 60 nt-long RNA. Upper inset shows SR linker (residues 183 to 195 in sticks) contact with RNA whereas the lower inset shows the interaction between NTD N-terminal tail (residues 26 to 42 in sticks) and RNA. One N protein monomer is colored in orange and the other one is colored in cyan. RNA is colored in yellow. Flexible regions of the NTD and CTD tails outside the density map was not presented for clarity. The complete fitted structure is presented in S4B Fig. C- The same N protein orientations as presented in B, but only showing structured regions (NTD and CTD) of N dimer without its non-structured regions (NTD tail, SR-linker and CTD tail). The two NTD units are represented above the CTD dimer. Upper inset shows, as spheres, positively charged residues of the CTD region (residues 248 to 280) that was previously demonstrated to be implicated in RNA binding [28,36]. Lower inset shows the superposition of the NTD from the structural model with the NMR NTD structure complexed with a 10-mer RNA in magenta (PDB code 7ACT [23]).

[47] performed with the 60 nt-long RNA and converted the CG models to all-atom models to select the structures that best fitted into the 27 Å resolution density map (accessed by the fourier-shell correlation in S5A Fig). This process of using a conformational sampling from CG simulations to correlate with experimental structural data was previously applied in [48,49]. We found that the NTDs are oriented side-by-side facing the CTD dimer (Figs 5B and S4B). All domains, including the flexible regions, contribute to shield the RNA segment. Interestingly, according to this model, there are no significant interface contacts between the CTD dimer and the NTDs (Fig 5B and 5C). As already mentioned, most class-1 particles display a weak density at the center, which indicates the existence of empty space between the globular domains (Fig 5A) and corroborates the idea that these domains do not fully interact with each other [50]. It is noticeable that, in the proposed model, the interaction of the RNA with the NTD resembles the binding mode of the NTD to a 10-mer RNA reported recently [23] (Fig 5C, lower inset). A similar N protein interaction region formed by the NTD finger-like hairpin was also reported for dsTRS and ssTRS [51]. Likewise, amino acid residues implicated in RNA binding in the CTD [17,22] are also involved in RNA interaction in our model (Fig 5C, upper inset). In addition, the intrinsically disordered regions represented by the N-terminal end and SR linker also make contacts with the RNA (Fig 5B). Interestingly, according to the model, the arginine and lysine residues of the SR linker interact with RNA while the serine residues, putative sites for phosphorylation, remain solvent-exposed and thus accessible to protein kinases (Fig 5B).

Together, molecular simulations and 3D reconstructions from NSEM images suggest that full-length N undergoes domain compaction in the presence of RNA and offer an interpretation of how the NTD and CTD pack together in the protein dimer.

## The octameric model of full-length N bound to RNA

In addition to the toroid-like particles described above, we inspected the class-2 square-like particles, because these particles were also commonly found in the RNA-containing N preparations and presented a clear organization pattern with four rounded units connected by the edges (Figs 3B and 6A).

To understand the structural organization of such particles in more detail, we collected ~25.000 class-2 particles and classified them into 150 subclasses (Fig 6A). These particles are ~13 nm wide and their rounded units located at the square vertices had dimensions varying from 5 to 7 nm. The size of these rounded units is consistent with the dimensions measured in the most populated class averages, the toroid-like particles (Fig 6A). The dynamic nature of the full-length N protein is highlighted by structural variations even within the same class (Fig 6A).

The 2D class average analysis also revealed particles showing a gap between two adjacent rounded units, like a U-shaped particle (Fig 6A, right top inset). Of notice, several square-like particles showed a blurred appendix of similar dimensions as the toroid-like particles (Fig 6A, right bottom inset). Other well-defined patterns comprise particles that seem to be composed of only three rounded units (Fig 6A, left top inset). These findings suggest that the square-like particles are formed by independent N protein units, probably dimers compatible with toroid-like particles. Moreover, the NSEM images illustrate the pleomorphic nature of the N protein oligomers. As non-identical particles are an obstacle to finely uncover the N protein structure through single particle averaging protocols, we used the CG analysis to generate 2D template reprojections. These simulated reprojections led a new NSEM image particle picking and rationalized a 3D reconstruction of the squared particles, providing a possible N protein structural organization of high order oligomerization.

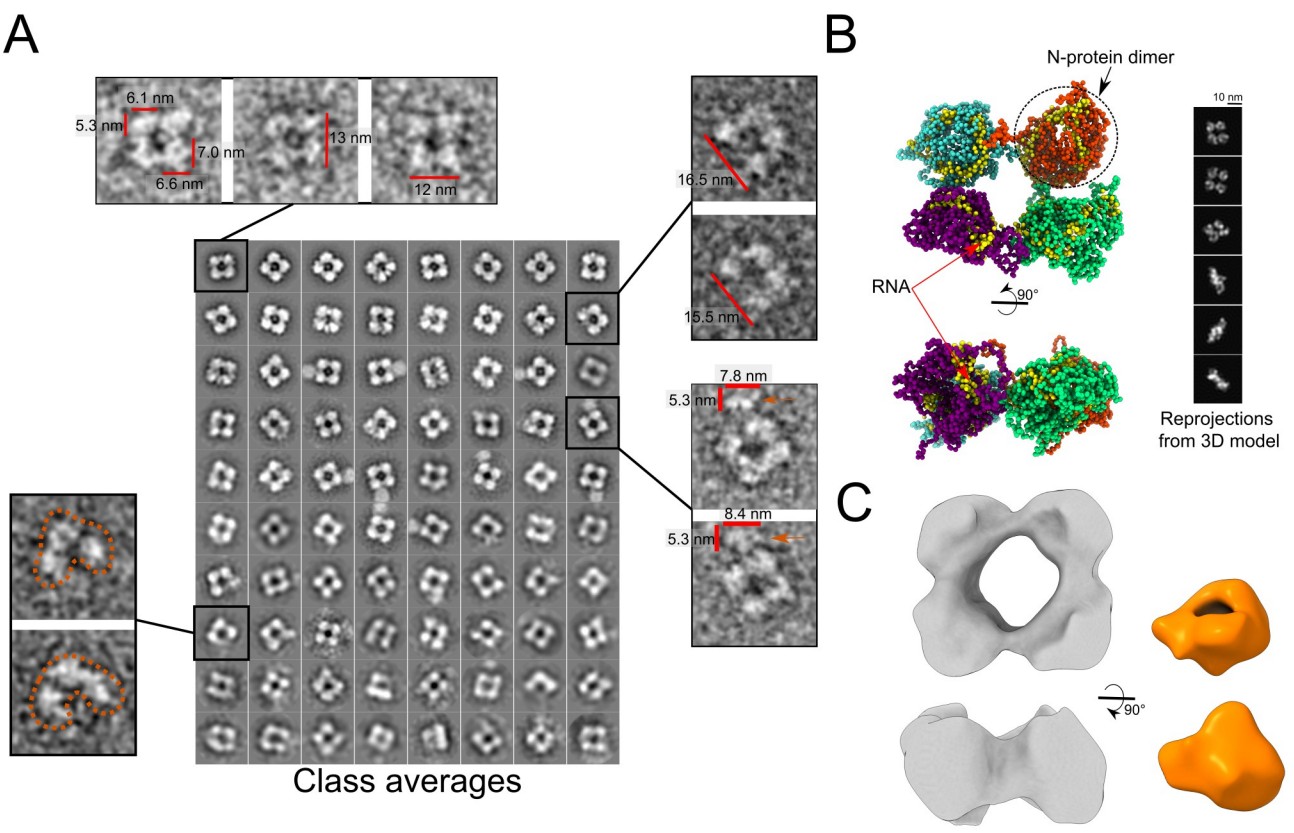

**Fig 6. 2D Classification and 3D density map reconstruction of the square-like particles.** A- 2D classification of square-like particles picked with xmipp3. Class averages are sorted by the number of class members (number increase up and left). Raw projections of particles that compose the classes are shown on insets. B- Last frame of N protein octamer CG simulation in the presence of 60-nt RNA for each dimeric unit. Reprojections from the 3D structure are shown on the right. C- 3D-density map of the square-like particles in comparison to the map of toroid-like particles, both rendered at 3-sigma contour level.

Thus, considering that each of the globular corners of the square-like particles has dimensions of 6 nm across and are formed by four independent N protein units, we modeled four copies (octamer) of the most packaged N-protein dimer from the 60-nt-long RNA simulation by placing each copy at the vertices of a virtual square. Then, in CG simulations, we connected each N protein dimeric unit through their C-terminal tails, based on the role of these tails in N protein oligomerization [25–27,52] (see next session and CG molecular dynamics simulations in Methods for details), and allowed approximation and accommodation of the oligomeric system (Fig 6B).

From the CG octameric structure, we built a simulated density map at 20 Å resolution, close to the resolution of the negative stain images, and generated reprojections from the 3D map at different orientations (Fig 6B). Remarkably, some reprojections are like the top view of the square-like particles and their dimensions. Further, reprojections corresponding to the side view of the octameric structure (Fig 6B) are consistent with the NSEM images of class-6 and class-7 particles (rod-like particles, Fig 3B). This finding was crucial because suggested that square and rod-like particles correspond to different orientations (top and side view, respectively) of the same particle in different orientations. Another possibility would be that rod-like particles correspond to dimer of dimers (tetramers) viewed along their plane. However, the size of longer rod-like particles (ranging from 180 to 190 Å) would not be compatible with the size of dimer of dimers viewed along their plane (~130 Å). Thus, by merging the

original square-like particles with the set of rod-like particles, we built a low-resolution (26 Å) 3D density map for the square-like particles (Figs 6C and S5). The 3D reprojections and original particle projections of both square and rod-like particles match appropriately, validating the reconstruction (S6 Fig).

This map revealed four quasi-globular units connected through the edges in a planar configuration (Fig 6C). The volume of each globular unit resembles the volume of the toroid-like particles (Fig 6C), thus reinforcing the idea that each globular unit contains an N dimer. This structural organization is similar to the RNP particles purified from the MHV3 and HCV229E strains observed by negative staining [44]. Taken together, our data suggest that the square-like particles observed in NSEM images represent the top views of octamers of the N protein.

## van der Waals forces guiding the C-terminal tail self-interaction

The ability of the N protein to form dimers, tetramers, and higher-order oligomers in solution has been reported previously [17,22,27,36]. Nevertheless, how exactly N dimers interact with each other to form such higher-order oligomers is presently unknown. The N protein octameric organization proposed in the section above is quite complex and despite our density map being very informative, this map did not inform atomic details of the interaction involving N protein dimers.

*In vitro* studies suggest that the C-terminal tail residues 343–402 in SARS-CoV-1 and 365–419 in SARS-CoV-2 are required for N protein tetramerization [25–27,52]. Considering the high degree of identity shared by these regions between SARS-CoV-1 and SARS-CoV-2, a more restricted dimer-dimer interaction zone, comprising residues 365 to 402, can be reasoned. This region (365 to 402) is predicted to be unstructured in all coronavirus N proteins (S7 Fig). Its adjacent C-terminal end residues, predicted to form an alpha-helix, are also conserved in most coronaviruses (S7 Fig). Thus, in order to identify potential protein-protein contacts involved in oligomerization, we performed all-atom molecular dynamics simulations of the C-terminal tail.

One μs simulations starting with separated C-terminal tail monomers, *a* and *b*, revealed that the two monomers adopt a similar folding, retaining two alpha helices named α1 (375–382) and α2 (400–419) (S8 Fig), consistent with the secondary structure prediction (S7 Fig). The simulations also revealed intrachain contacts in each C-terminal tail monomers (Fig 7A and 7B). *Monomer a* showed persistent van der Waals interactions between residues 407–415 (helix α2) and residues 377–385 (helix α1) and its neighbor coil residues L382 and R385 (Fig 7A and 7D). The contacts involving helix α2 (N406, L407, S410, M411, S413, A414) with the coiled region of *Monomer b* (mainly residues P383, V392, L395) were also observed (Fig 7B and 7E). Conformational ensembles were obtained by clustering the simulation trajectory into six groups (covering ~70% of the analyzed frames). The ensembles evidenced that the C-terminal tails complexes were well-packaged with a stable dimeric interface surrounded by flexible regions (S9 Fig). The most frequent interface contacts involved residues 394 to 398 from one monomer with 405 to 412 from the other (Fig 7C and 7F). The residues 394 to 398 correspond to a hydrophobic sequence LLPAA and an energy contribution analysis suggested that van der Waals interactions are the major forces driving the C-terminal tail dimerization (S10 Fig). The conformational ensembles, including the most populated one (cluster 2), evidenced the participation of both helices α2 in the C-terminal tail interface (S9 Fig).

## Discussion

The underlying mechanism of how the N protein associates with the RNA to form the RNP particles remains unknown. Likewise, the molecular basis of the N protein self-association is

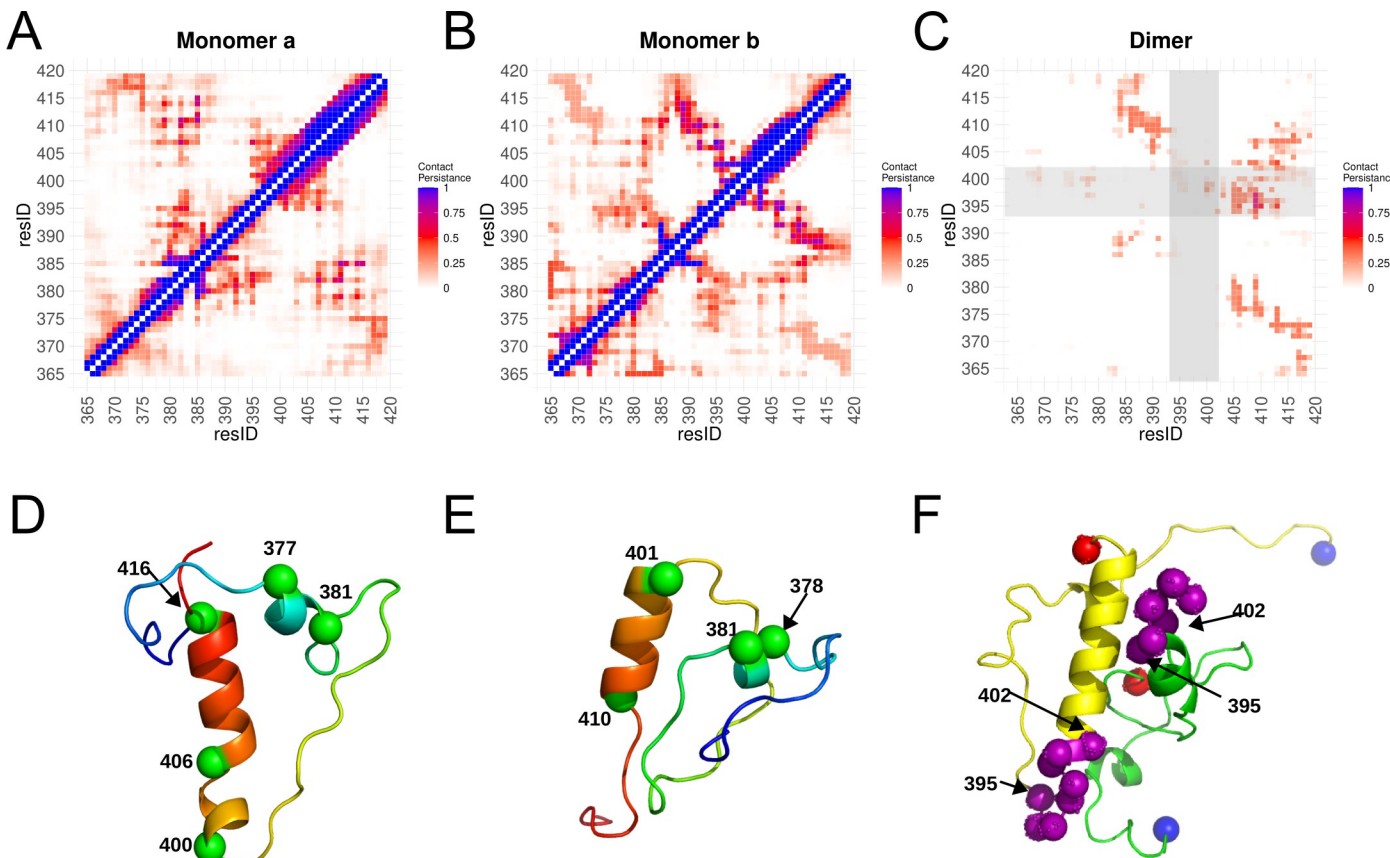

**Fig 7. All-atom molecular dynamics simulation of the N protein C-terminal tail.** A- and B- Contact maps for each individual monomer (a or b) and C- for C-terminal tail dimer. D-, E- and F- Representative structures of contact maps. The monomer contact maps (a or b) is an average of the whole trajectory of five MDs, considering only the last 50 ns of each trajectory for making the dimer contact map. The scale is defined as the contact persistence from 0 (none) to 1 (along all the simulation). D- and E- Structure of each monomer colored from N-terminal in red to C-terminal in blue, some Cα atoms in green VDW representation as reference of the residue numbers. F- A representative structure of the dimer. Each monomer is in new cartoon representation, one in yellow and the another in green. In purple, the hydrogen atoms from the residues 395 to 402 suggested to participate in the dimer interface (grey transparent regions in C, see [27]). Spheres in blue and in red indicate the N- and C-terminal of the C-terminal tail used in the simulation.

also poorly understood. The understanding of these processes, which are fundamental for the virus life cycle, has been hampered by the dynamic structural nature of the N protein. Thus, despite the wealth of structural data for the NTD and CTD regions, no structural models are available for the full-length N protein. Here, by combining NSEM with biochemical/biophysical and *in silico* approaches, we propose structural models and their dynamics to explain the manner of the dimer of the full-length N protein self-organizes both in the absence or presence of RNA. We also proposed a model of full-length N bound to RNA in an octameric organization that is reminiscent of RNP particles isolated from other coronaviruses [40,42].

The negative stain images of full-length untreated N bound to bacterial RNA provided an informative panel of N protein structural organization in the presence of large variety of nucleic acids. This is quite relevant because the structural organization of SARS-CoV-2 N protein is still poorly understood and only few studies have addressed this issue. The negative stain images showed a wide range of particles of different sizes and shapes, including large helical-like structures, which reflected the pleomorphic nature of this protein. Although particles of N protein incubated with polyA15 has a globular shape consistent with class 1 particles, thought to contain *E. coli* RNA, those derived from samples incubated with polyA50 showed irregular shape and were distinct from the well-organized class 2 particles, which are also

believed to carry bacterial RNA. This reflected the dynamics of the N protein upon RNA binding and suggests that sequence-specific RNAs from *E. coli* can drive the packing of the N protein more effectively than the polyA molecules tested.

Despite N protein being typically described to form phase-separation [28,29], experimental conditions seem to be critical to induce such process. Here, the N protein preparations, even in the presence of RNA, did not present any evidence of phase-separation. The absence of phase-separation can be consequence of N protein concentration, which here is below the concentration demonstrated to induce phase-separation. Likewise, the RNA lengths used here (polyA15 or polyA50) are shorter than the one used in experiments to evaluate phase-separation [29].

Previous electron microscopy studies have suggested that the RNPs isolated from MHV and SARS-CoV-1 virions display a helical structure [41,42]. Furthermore, two studies using cryo-electron tomography have proposed that native SARS-CoV-2 RNPs are highly heterogeneous and densely packed but locally ordered in the virus, with neighboring RNP units with dimensions about 14 nm organized in a "beads on a string" fashion [43,53]. Moreover, although further studies are still required to confirm a helical organization of the SARS-CoV-2 RNP, the coiled structures of the MHV RNPs of about 11 nm in diameter with a 4 nm space [41] are quite compatible with the dimensions of the square-like (class 2) particles reported here. To our knowledge, these square-like particles have not been described for SARS-CoV-2 or any other coronaviruses N protein, although they are remarkably similar to electron microscopy images of RNPs isolated from MHV3 and HCV-229E [44]. We thus propose that the N protein octamers with the shape of the square-like particles described here could represent building blocks of the SARS-CoV-2 RNP structure, as suggested for the MHV RNP [42].

To form such higher-order structures with the RNA, the N protein is thought to also depend on protein-protein interfaces. Accordingly, the C-terminal tail of the coronavirus N proteins has been implicated in protein tetramerization [25–27,52]. Recently, C-terminal tails were reported to be crucial for the interaction between N protein dimers, and their deletion was sufficient to disrupt phase-separation [54]. Here, using molecular dynamics, we investigated how the C-terminal tail could play a role in the N protein oligomerization. We found that the C-terminal tail adopts a folded structure maintained by van der Waals intrachain contacts involving two helices. Moreover, the hydrophobic segment LLPAA appears to play an important role in tail-tail interaction, which agrees with HDX values for this segment reported previously (Fig 7F) [27]. The hydrophobic character of the C-terminal tail is thus thought to drive the formation of higher order oligomers (tetramer and octamers) of the N protein. A similar mechanism was found in Tula hantavirus N protein, in which the hydrophobic IILLF segment located at the C-terminal tail of its N protein was essential for the protein oligomerization [55]. The relevance of the C-terminal tail as a target to affect N protein oligomerization was investigated in the related coronavirus HCoV-229E, where a tail-derived peptide was shown to reduce the ability of N protein to form high-order oligomers [26]. In this sense, peptidomimetic molecules could thus be designed and used to disrupt this tail-tail interface to prevent N protein oligomerization.

The dynamic nature of N as an RNA-binding protein was also revealed by CG simulation models, which corroborated the NSEM images and Rg of the protein in the absence and presence of RNA [19,36]. According to these models, in the absence of RNA, the N protein oligomerizes into dimers where the NTDs move freely relative to the CTD dimer, as previously suggested [36,56]. Conversely, in the presence of RNA, the protein undergoes a compaction that brings the NTDs closer to the CTD dimer. This compaction was confirmed by DLS measurements of the protein with polyA15. Despite the proximity of the NTD and CTD upon RNA binding, we did not observe a direct contact between these domains, as suggested by the

weak density at the center of the 3D density map of the toroid-like particles, which resemble MHV dimeric N protein particles [42]. The CG simulations suggested multiple RNA-binding sites in the N protein dimer and how the RNA molecules interacted with the structured domains is also consistent with the RNA-binding mode determined experimentally for NTD [23], and with the RNA-binding surface observed in the CTD [17,22]. The CG models also suggested a role of the SR-linker in RNA binding, which is in line with previous findings [9,14,19]. For instance, the highest frequency of RNA contacts observed in SR-linker during CG simulations occurs in the region of R189 residue, which was previously proposed to play a critical role in RNA binding [29]. In addition, mutations that enhance the positive charges in the SR-linker, such as the G204R mutation found in one of the high-transmissible SARS-CoV-2 variant [31], could alter the N protein affinity to the viral RNA. Considering that phosphorylation of the SR-linker plays a key role in the coronavirus life cycle [34], it is noteworthy that, in our CG models, the positively charged residues of the SR-linker point towards the RNA, while the serine residues are solvent-exposed and thus prone to be phosphorylated by protein kinases. It is important to note that in the context of infected cells, the RNA binding profile could be affected by N protein binding proteins, such as nsp3, as evidenced by [56].

In addition, the N protein dimer models provided here offer further insights into how a single protein dimer interacts with a single RNA molecule of varying lengths without considering higher-order oligomerization. These models aimed to simulate how the N protein binds to the viral genome, where several protein dimers are thought to compete for a short genome segment. Our CG models suggested that a genome segment ranging from 40–60 nt would be sufficient to occupy all the RNA-binding sites and induce protein compaction. However, such RNA-binding sites could be simultaneously occupied by multiple short RNA segments, as suggested by the GC models and confirmed experimentally with polyA15 and polyT10 [37]. Likewise, longer RNA segments could drive protein oligomerization, as observed with polyA50 and polyT20 [37]. In this sense, despite the density map of the octameric N protein organization not revealing atomic details, it could be possible that RNA participates in the interactions between dimers within the octamer, in addition to the CTD-tail interaction.

In conclusion, our results shed new light on the dynamics and higher-order oligomeric structure of the SARS-CoV-2 N protein and provide a framework for understanding the multifunctional and versatile role of this protein.

## Methods

### Cloning procedures

The SARS-CoV-2 RNA was isolated from virus particles with the QIAmp viral RNA mini kit (Qiagen—USA) and reversely transcribed to cDNA with the High-Capacity Reverse Transcription Kit (Thermo—USA). The N protein sequence (GenBank QIG56001.1) was amplified from cDNA samples using primers SC2-protN28182-F (5'-AGTCTTGTAGTGCGTTGT TCG-3') and SC2-protN29566-R (5'-ATAGCCCATCTGCCTTGTGT-3') and cloned into pGEM-T Easy (PROMEGA—USA), generating plasmid pGEM-SC2-N. The N sequence was reamplified from pGEM-SC2N with forward 5'-AACAAGCTAGCATGTCTGATAATGG ACCCCAAAATCAG-3' and reverse 5'-GGTCTGCGGCCGCTTAGGCCTGAGTTGAGTC AGCACTGCT-3' primers and subcloned into the *Nhe*I/*Not*I sites of a pET28a-TEV vector carrying a 6xHis-tag and TEV protease cleavage site at the N-terminus.

### Protein expression and purification

The N protein was expressed in *E. coli* BL21 (DE3) cells (Novagen -USA) and purified by metal-affinity and SEC. Cells were grown at 37˚C under agitation (200 rpm) in LB medium

containing kanamycin (50 mg/L) to an optical density (OD600nm) of 0.8. Recombinant protein expression was induced by the addition of 0.1 mM isopropyl-thio-β-d-galactopyranoside (IPTG) for 16 h at 25˚C. After centrifugation, cells were resuspended in 50 mM sodium phosphate, pH 7.6, 300 mM NaCl, 10% v/v glycerol, 1 mM phenylmethylsulfonyl fluoride and incubated on ice with lysozyme (0.1 mg/mL) for 30 min. Bacterial cells were disrupted by sonication and the soluble fraction was loaded on a 5 mL HiTrap Chelating HP column (GE Healthcare—USA) previously equilibrated with same buffer. Proteins were eluted using a linear gradient (20 to 500 mM) of imidazole at a flow rate of 1 mL/min. Eluted fractions containing the N protein were pooled, concentrated and loaded on a HiLoad 16/60 Superdex 200 column (GE Healthcare), previously equilibrated with 10 mM Tris, pH 8.0, 100 mM NaCl, at a flow rate of 0.5 mL/min. These samples, still carrying bacterial RNA, were named as untreated N protein samples in the text.

To produce the N protein without nucleic acid contaminants we treated the purified N protein with RNAse A or we purified N protein in the presence of urea and high salt concentration. To treat N protein with RNAse A, purified N protein was incubated with RNAse A (Qiagen) with 1:15 (RNAse A: protein) proportion for 1 h at room temperature. After incubation, the samples were loaded on a 16/60 Superdex 200 column equilibrated with 50 mM sodium phosphate, pH 7.6 and 500 mM NaCl. Eluted fractions containing the N protein were pooled and concentrated.

To produce urea-treated N protein samples, *E. coli* cells were lysed in buffer A (50 mM sodium phosphate, pH 7.6, 500 mM NaCl, 20 mM imidazole, 6 M urea, 10% glycerol). The suspension was sonicated and centrifuged at 18,000 x g for 45 min at 4˚C. The supernatant was applied on a HiTrap Chelating HP column equilibrated with the same buffer. Proteins were eluted with a linear imidazole gradient using buffer B (50 mM sodium phosphate, pH 7.6, 500 mM NaCl, 300 mM imidazole, 3 M urea, glycerol 10%). Protein fractions were mixed and dialyzed against buffer C (50 mM sodium phosphate, pH 7.6, 500 mM NaCl, 10% glycerol). Protein samples were centrifuged at 14000 x g for 10 min at 4˚C and subjected to molecular exclusion chromatography on a Superdex 200 16/60 column, equilibrated in buffer C, under a flow rate of 0.7 mL/min. Protein purity was analyzed by SDS-PAGE and protein concentration was determined by absorbance at 280 nm using the molar extinction coefficient calculated from the amino acid composition. Protein samples were concentrated and stored at– 80˚C. In experiments involving polyA15 or polyA50, except for SEC-MALS analysis, urea-treated samples were diluted 5x (DLS) or 20x (NSEM) in 50 mM sodium phosphate, pH 7.6 buffer (without NaCl) prior to RNA incubation.

## Circular dichroism analysis

Protein samples at 6 to 30 μM final concentration were diluted in 50 mM sodium phosphate buffer, pH 7.6, and analyzed by FAR-UV circular dichroism. All measurements were recorded on a Jasco J-810 Spectropolarimeter at 10˚C, in the range of 197–260 nm. After buffer signal subtraction, CD signals were normalized to residual molar ellipticity using the equation $\theta = (mdeg.100.MW)/ (mg/mL.l.NR)$, where mdeg = CD signal, MW = protein molecular weight, mg/mL = protein concentration in mg/mL, l = optical path in centimeters and NR = protein residues number.

## Dynamic light scattering

N protein samples (~ 20 μM) in 50 mM sodium phosphate, pH 7.6, 100 mM NaCl, 2% glycerol, were submitted to dynamic light scattering analysis in the ZetaSizer NanoZS90 (Malvern) equipment. All the measurements were acquired following the equipment automatic setup,

and after sample equilibrium at 10–18˚C. All buffer parameters corrections were performed by Zetasizer Nano software. Obtained data are shown as the average of at least three measurements. The statistical approach used to compare the conditions was performed in R with DTK R package that applies the Dunnett-Tukey-Kramer pairwise multiple comparison test adjusted for unequal variances and unequal sample sizes. The P value was obtained from the confident interval as described in [57].

## SEC-MALS analysis

N protein samples (~ 20 μM) were loaded onto a Superdex 200 10/30 column, equilibrated with 50 mM sodium phosphate, pH 7.6, 500 mM NaCl, under a flow rate of 0.4 mL/min. To inspect the RNA influence on the N protein oligomeric state, samples of N protein in 5-fold excess of RNA polyA50 or polyA15 were also analyzed following the same protocol. For this, the mixture (protein-RNA) was incubated, on ice, for 1h, before the injection. SEC-MALS analyses were performed on a Viscotek OmniSEC (Malvern, UK) equipped with a SEC module coupled to a two-angle laser light scattering detectors, a refractometer and a viscometer. The OmniSEC software was used to acquire and evaluate the data. All the graphs were obtained using OriginPro 2021 software.

## Negative staining and imaging

To collect negative stain images of the untreated N protein, 3 μL of purified N (4 μM) in 10 mM TRIS buffer, pH 8.0, 100 mM NaCl were applied onto glow-discharged (15 mA, negative charge for 25 s) 400-mesh copper grids covered with a thin layer of continuous carbon film (01824—Ted Pella, USA). After 1 min, the excess liquid was drained using a filter paper. The grids were stained twice with 3 μL uranyl acetate solution (2%) for 30 s. The excess solution was drained and the grids were allowed to dry at room temperature. Automated data collection was performed using a 200 kV Talos Arctica G2 transmission electron microscope (Thermo Fisher Scientific). Data set of 18148 micrographs were automatically acquired with EPU software and recorded on a Ceta 16M detector. The pixel size and defocus were 1.96 Å and -1.5 μm, respectively. The exposure time was 1 s in an accumulated dose of ~23 e-/Å$^2$.

To collect negative stain images of N protein treated with urea in the absence or the presence of polyA15 or polya50 we used the same grid preparation protocol described above. Urea-treated samples in 50 mM sodium phosphate, pH 7.6, 500 mM NaCl, 10% glycerol were diluted 20x in the same buffer but without NaCl before incubation for 2 h at room temperature with RNA. The protein final concentration was 0.5 μM and we used 1:10 protein-RNA molar ratio in N protein samples with RNA. Screening data was performed using a 120kV JEM-1400Plus transmission electron microscope (JEOL, Japan), equipped with OneView 16-Megapixel Camera (Gatan, USA), magnification of 80k and pixel size of 1.39–1.89Å.

## Image processing

A total of 17370 collected negative stain micrographs (4096 x 4096) from purified N protein (untreated-N) preparations were preprocessed using Imagic-4d software system [58]. Raw micrographs were firstly submitted to *a posteriori* camera correction and then to a contrast transfer function (CTF) correction. For CTF correction, the amplitude spectrum of each image was determined and submitted to eigenvector analysis and automatic unsupervised classification into 2000 classes. The resulting class averages were used to determine the CTF parameters, which were passed to the individual images. Then, the CTF correction was applied by phase-flipping each image. A total of 15474 CTF-corrected micrographs were selected

based on the quality of CTF estimation and defocus for further processing. All image processing programs, except Imagic, were run in the Scipion framework [59].

Micrographs were resized in Fourier space to 1024 x 1024 dimensions and submitted to particle picking using the Xmipp3 package [60]. The picking was performed in original 4096 x 4096 size, by an initial manual picking of ~550 particles from 40 micrographs, and proceeded by automatic picking, resulting in 178,021 particles in a 200 x 200 box size. The particles were subjected to a round of 2D classification in Relion to separate toroid, square and rod-like particles groups. A total of 24137 toroid-like particles (class 1), 33550 square-like particles (class 2) and 12552 rod-like particles (classes 6 and 7) were used for further processing.

Given the structural pleomorphism of purified N protein, we cautiously evaluate the use of classical single-particle protocols. Toroid-like particles were classified into 300 class averages using Relion [61] and an initial 3D model (C1 symmetry) was built using 30 class averages in Eman2 [62]. Class averages placed at a 112 x 112 box with a circular soft mask (0.52 radius and 0.05 drop-off) were band-pass-filtered (0.001 low pass and 0.1 high pass). Then, the initial 3D map was refined in Imagic using an iterative process of angular reconstitution and class average rotation and translation alignment to 3D reprojections, achieving 3D resolution convergence.

In the case of square and rod-like particles, these particles were cautiously merged into a combined particles dataset being informed by the 3D structural models obtained from CG molecular dynamics simulations of N protein octamer. From these particles, an *ab initio* reference-free initial volume was generated in Relion, followed by two independent rounds of 3D classification. We calculated the consensus [63] of both 3D classifications (resulting in a total of 9 3D classes) and selected a single stable 3D class corresponding to 17,803 particles (38.6% of the particle set). This 3D class was further auto-refined following the 0.143 FSC target in Relion, obtaining self-consistent convergence in 13 iterations. The 3D density unmasked map resolution was accessed by the Fourier Shell Correlation (FSC), using ½-bit threshold (S5 Fig).

## CG molecular dynamics simulations

CG molecular dynamics simulations were performed using CafeMol 3.1 software [64]. An initial N protein dimer all-atom model was built in YASARA software [65] using crystallographic structures of NTD monomer (PDB ID: 6VYO) and CTD dimer (PDB ID: 6WJI). For this, NTD monomers were placed at a distance that allowed a fully extended NTD-CTD linker conformation. Intrinsically disorders domains (N-terminal tail, NTD-CTD linker and C-terminal tail) were modeled using YASARA. The modelling was based on the N protein sequence from GenBank QIG56001.1, the same used in experimental assays. For simulations in the presence RNA, an initial single strand RNA all-atom model was modeled without tertiary structure or base pairing in YASARA. Nucleotide sequence of all RNA of the same size used in simulations were the same and consist of a scrambled sequence, since the CG model used does not compute base specific interactions.

Five independent simulations with different random seeds were conducted for each dimeric condition (without RNA or in the presence of a single RNA segment of 10 to 70 nucleotides) by Langevin simulations. For each independent simulation, the RNA was initially placed at different aleatory positions in relation to the protein. The temperature was set to 300 K. Local protein-protein and RNA-RNA interactions were modeled using AICG2+ and GO potentials [64], respectively. For non-specific interactions between N protein monomers, and between N protein monomers and RNA, excluded volume and electrostatic interactions were considered. Electrostatic forces were computed using Debye–Hückel equation with a cut-off of 20 Å and ion strength of 0.1 M. To positively charged residues +1e was assign whereas for negatively charged

−1e. Local GO potential was changed to Flexible Local Potential in intrinsically disorder regions: NTD tail, SR-linker and CTD-tail. Each simulation was run for $1.5 \times 10^7$ steps, with a time length of 0.2 for each step in CafeMol scale. For dimers simulations, only the last $5 \times 10^6$ were used in MD analysis. The convergence of CG simulations was accessed by the RMSD calculation. The entire trajectory was aligned by the CTD atoms of the N protein dimer using the first simulations frame as reference and the RMSD was calculated using all the N protein dimer atoms (S12 Fig).

For N protein octamer simulation, we built an initial model using four N protein dimer copies from the last all-atom fitting model. Each dimer was placed 200 Å from the center of mass of the neighbor dimer. In CafeMol simulations, a harmonic spring (force coefficient of 0.005 and distance constraint of 5) was used to bring together C-terminal tails from neighbor dimers. The contacts involving each dimer was modeled using local GO potential to maintain native structure of both protein and RNA. In addition to the harmonic spring, we used excluded volume and electrostatic interactions to model the contacts among the four dimeric units. The octamer simulation was run for $5 \times 10^6$ steps, which was sufficient to stabilize the RMSD calculated from CA atoms. The last frame from the simulation was further analyzed. The convergence of the CG simulation was accessed by the RMSD calculation. The entire trajectory was aligned by all atoms of N protein octamer using the first frame as reference and the RMSD calculation was performed using N protein octamer atoms (S13 Fig).

## Tools for comparing CG simulations with biophysical or electron microscopy data

For comparing CG simulation Rg with biophysical data, we estimated the Rg of the simulated systems with Bio3D package using an in-house R script [66].

For comparing 3D atomic structures or CG models with negative stain images, we reprojected the structures or the models into 2D simulated images. Then, we estimated the area of these simulated reprojections as well as the area of the particles from negative stain images. For the full-length N protein dimer, we used 3 structures obtained from CG simulations: structures with the minimum, mean and maximum Rg along the simulation. Prior to reproject, CG structures were converted to all-atom structures. For NTD monomer and CTD dimer structured domains, we used crystallographic structures of these domains obtained from Protein Data Bank (PDB ID: 6M3M [24] and 6WZO [27], respectively). All the structures, obtained from CG simulations or PDB, were used to build 20 Å resolution simulated density maps with Chimera [67] using *molmap* function. Then, the 3D density maps were reprojected in different orientations (rotational angle 0 to 360 degrees with step of 20 degrees and tilt angle 0 to 180 degrees with step of 20 degrees) with Xmipp3 *create gallery* function in Scipion workflow. A total of 100 2D reprojected images were used for each structure and, before any comparison, images were resized to match the pixel size values.

To estimate the area of simulated reprojections of 3D structures or the area of the particles from the negative stain images. we used ImageJ software [68]. In ImageJ software, we first adjusted a threshold to indicate the region of the image corresponding to N protein, as showed in S2B Fig. In the case of the full-length N protein dimer from CG simulations, its simulated reprojection can be seen as three high contrast regions, depending on its orientation and conformation (S2B Fig). These regions correspond to the N protein structured domains (NTD and CTD), and in this case, these regions were computed separately in the area estimation process.

## 3D atomic model fitting

From the previous independent simulation of the CG model of N protein dimer in complex with a 60-nt-long RNA were selected 100 structures with the lowest Rg using only the

structured domains (NTDs and CTDs). The correlated structures were removed by clustering. The structures of the reduced set (~40 structures) were converted to all-atom structure model. The protein was reconstructed using PULCHRA [69] and the RNA was converted into all-atom model using DNAbackmap tool from CafeMol.

The structures were rigid-body docked on the density map by employing *colores* program from Situs package [70]. Three different docking calculations were performed by selecting three different structural segments of the system: 1) NTDs, CTDs and RNA heavy atoms; 2) NTDs and CTDs heavy atoms, and 3) all heavy. The docked structures at each group were ranked by its cross-correlation coefficient (CCC). The selected structure for the model fitting was that with the lowest CCC and with close spatial coordinates in all the docking calculation groups. A compatible resolution with the density map (25 Å) was used for the conversion from PDB to volumetric map.

The VMD AutoPSF plugin was used to generate a topology for MD simulations and MDFF simulations. The complete all-atom model (protein-RNA) was solvated in a TIP3P water box of 134x191x163 Å$^3$ using Solvate plugin from VMD. Na$^+$ and Cl$^-$ ions were added to neutralized and to adjust the ionic strength to 150 mM using the VMD plugin autoionize [71]. The complete system involved 396,013 atoms. The molecular interactions were described by CHARMM36 force field with CMAP corrections [72].

All simulations were performed with NAMD 2.13[73]. The system was equilibrated using the following protocol: 1) 10,000 minimization steps of water molecules and ions by restraining the protein and RNA; 2) 10,000 minimization steps of the complete system; 3) 200 ps of NVT MD fixing positions of the protein and the RNA and 4) 200 ps of NPT MD fixing positions of the protein and the RNA. The temperature was maintained in 300K using a Langevin thermostat with damping coefficient of 1 ps$^{-1}$ and coupling only heavy atoms. In the NPT simulations, the pressure was maintained constant in 1 atm using Nose-Hoover barostat with a piston decay of 200 fs and a piston period of 400 fs. Particle Mesh Ewald (PME) method were applied for describing long-range electrostatic interactions [74]. A non-bonded cut-off of 10 Å was applied to calculate electrostatic and van der Waals potential (vdW). A shifting function was applied to electrostatic interactions for avoid truncation of the potential function. vdW interactions were smoothed by applying a switch function at distance of 9 Å. The time step was 2 fs using SHAKE for all covalent hydrogen bonds of the protein and nucleic molecules [75] and SETTLE algorithm for rigid water molecules [76].

MD Flexible Fitting of the protein-RNA into the density map were performed following a multi-step protocol [77]. The protocol involved three steps: 1) 5 ns MDFF simulation, applying a scaling factor used in the energy potential of the map ($\xi = 0.3$ kcal mol$^{-1}$) on all heavy-atoms of the RNA and NTDs and CTDs. Also, the dihedral angles of α-helices and β-sheet of the NTDs and CTDs were constraint using harmonic restrains (k$_{\mu,protein}$ = 400 kcal mol$^{-1}$ rad$^{-2}$). 2) 5 ns MDFF using similar conditions of the step 1, but including the all heavy-atoms of the intrinsically disordered regions (N- and C-terminal tails and SR-linker) in the potential energy function of the map. 3) a minimization of 10,000 steepest descent steps, just restraining initial positions of the backbone protein and the RNA atoms with a force constant of 10 kcal mol$^{-1}$ Å$^{-2}$. 4) a minimization of 10,000 steepest descent steps without any restraint. The time step was 1 fs. SETTLE algorithm was used for rigid water molecules [76]. Each step was done until reach convergence of the cross-correlation coefficient between model and map, and convergence of the RMSD calculated from protein and RNA backbone (S11 Fig).

## All-atom molecular dynamics simulations

For C-terminal tail simulations, a folded linear protein (C-terminal tail involves residues 365 to 419) was built with YASARA [65], from PSIPRED secondary structure prediction [78]. This

structure shows one helix from residues 403 to 419 and was used as starting point for MD simulation of one C-terminal tail monomer. The MD simulation of the monomer was to check the folding along the trajectory without the influence of another monomer. The initial secondary structure did not change along the trajectory and no folding was observed. When the trajectory is aligned using the helix as reference, the coil region moves randomly. After a clustering, the average structure of the simulation was used as starting point for building a dimer system with the monomers separated by ~70 Å using Packmol program [79]. The large distance was set in order to decrease any possible bias on the starting configuration.

The dimer was immersed into an octahedral box with 56,098 TIP3P water box, 150 Na$^+$ ions and 152 Cl$^-$ ions. The dimer formation was studied by running five replicates. However, this system was described by ~170,000 atoms, needing an important computational effort to perform a 1 µs MD simulation. Thus, five uncorrelated starting points of the dimer were picked from the dimer MD simulation. The criterion was to select those with a minimum distance of 30 Å between any atom of the monomers. All the selected dimers were solvated again in octahedral box with 45,556 TIP3P waters, 121 Na+ ions and 123 Cl- ions in order to keep the system composition and the ionic strength (150 mM). All the systems were described using Amber ff14SB force field [80] and the topologies were generated using *tLeap* program from AmberTool20 [81].

All the MD simulations were equilibrated by the following steps: 1) minimization of the solvent by 2500 steepest descent steps followed by 2,500 conjugate gradient steps; 2) minimization of the whole system by 2,500 steepest descent steps followed by 2500 conjugate gradient steps; 3) the system was heated from 0 to 300 K in 200 ps under NVT conditions, restraining the protein atom positions; 4) the density was equilibrated under 500 ps under NPT conditions, restraining the protein atom positions. The production step of all the simulation was run for 1 µs under NPT conditions. For all the simulations, the temperature was set to 300 K using a Langevin thermostat with a collision frequency of 5 ps$^{-1}$. In the case of the NPT simulations, the pressure was set to 1 atm using a Monte Carlo barostat with pressure relaxation time of 1 ps$^{-1}$. The long-electrostatic interactions were calculated using Particle Mesh Ewald method [74] with a cut-off of 9 Å. SHAKE algorithm was applied to all bonds, allowing a time-step of 2 fs. The simulations were run using GPU accelerated PMEMD program that is part of the AMBER18 package [82–84].

The contact map analysis was done using *nativecontacts* command using two masks, one of each monomer, including all atoms and a cut-off distance of 7 Å. The contact maps figures were done using Rstudio[85]. The secondary structure along trajectory was performed using *secstruct* command. The inter-monomers vdW and electrostatic interaction energy was calculated using *lie* command with a dielectric constant of 78. All the commands are part of Cpptraj program from AmberTools20[81]. The images of the C-terminal tail monomers and dimers were generated using pymol 2.3.0 (open-source build). The convergence of the all-atom simulations was accessed by RMSD calculations. The trajectories were aligned by the backbone atoms of the two CTD-tail monomers using the first frame as reference and the RMSD calculation was performed using the same atoms selection (S14 Fig).

To generate a conformational ensemble of the quintuplicate CTD-tail trajectory (~25000 frames), we first estimated the radius of gyration using rgyr function from Bio3D R package [66] to remove unbound complexes (S9A Fig). Then, the bound complexes (17600 structures) were aligned, using the α-helix comprising residues 400 to 411 from one CTD-tail monomer as a reference, and the atomic coordinates were submitted to a principal component analysis (PCA), using the Bio3D R package. The two principal components (PC1 and PC2 in S9B Fig) were submitted to a density-based spatial clustering analysis with DBSCAN method [86] using the dbscan R package [87] with eps = 15 and number of minimum points = 100 (S10B Fig).

## Supporting information

**S1 Fig. Contact map of the N protein dimer in the CG simulations without RNA and the electrostatic surface potential of the N protein structure.** A- Contact map of the N protein dimer in RNA-free CG simulation. Contact map was built using Bio3D packaged. Five independent CG simulation runs were considered during frequency contact calculation and a cut-off distance between CG Cα beads was set to 10 Å. Relevant contact regions are indicated. B-Electrostatic potential surface of an extended conformation of the N protein dimer model. (TIF)

**S2 Fig. Representative NSEM images of the N protein in the absence or presence of RNA and the area estimation of particles.** A- Representative micrographs for each condition: urea-treated N protein in absence of RNA (left panel), or in the presence of polyA15 (middle panel) or polyA50 (right panel) at a 1:10 protein-RNA ratio. Scale bars are indicated. B–Representative 2D simulated reprojections in three different orientations of NTD monomeric crystallographic structure obtained from PDB 6M3M (upper panel), CTD dimeric crystallographic structure obtained from PDB 6M3M (middle panel) and full-length N protein structured obtained from CG simulations (bottom panel). Below each reprojections are presented the threshold applied to identify the high-density regions in images, which corresponds to the structured domains of N protein. In full-length N protein reprojections, under the threshold used we did not include the flexible regions of N protein that has low intensity value. C–Area distribution of 3D structures (NTD monomer and CTD dimer) or 3D model reprojections (full-length dimer) and class averages of particles collected from negative stain images of urea-treated N samples, same as in Fig 2B. Here, the area distribution of urea-treated N with polyA15 samples, were included in the comparison presented in Fig 2B of the manuscript. The area distribution of urea-treated N overlaps the area distribution of NTD and CTD reprojections, indicating that the particles observed in the negative stain images corresponds to these individual domains. The presence of polyA15 shifted the area distribution of urea-treated N, indicating that the particles have larger dimensions than without RNA, but still compatible with the larger dimensions observed in full-length dimer reprojections. This finding suggests that in presence of polyA15, the individual NTD and CTD domains got closer and the individual density regions that are separated, got together and formed a unique particle with larger dimensions in comparison to the individual domains. A 2D classification of the N protein + polyA50 particles could not be performed because these particles were not well delimited. (TIF)

**S3 Fig. CG dynamics simulations of full-length N dimers in the presence of 15-nt RNA molecules.** A- Density plot of the gyration radius distribution (in nm) calculated from five independent CG simulations of the N protein dimers in the presence of one, two or three single-strands 15-nt RNA. For comparison, we also presented distributions of conditions without RNA (0 nt) and in the presence of 60 nt long. B- Representative frame with minimum radius of gyration of full-length N dimer in complex with three 15-nt RNA molecules. N protein monomers are colored in red and green, whereas the RNA molecules are colored in yellow. (TIF)

**S4 Fig. Comparison between 2D particle projections and 2D reprojections of the 3D density map, and the N protein dimer model fitted into the 3D density map.** (A) Comparison between toroid-like class averages (even columns) and their reprojections from the 3D volume obtained in EMAN2 (odd columns) to validate the 3D reconstruction. (B) On the left, two different views of the 3D density map reconstruction of the toroid-like particles rendered using 3-sigma contour level. On the right, the same density map with the flexible fitting of the N

dimer atomic model derived from the CG simulations performed with the 60 nt-long RNA. All N protein domains are presented. The density map did not present density sufficient for the fitting of N protein CTD C-terminal tails or NTD N-terminal tails, probably due to their high mobility of these domains. One N protein monomer is colored in orange and the other one is colored in cyan. RNA is colored in yellow.
(TIF)

**S5 Fig. Fourier shell correlation profile of 3D density maps.** Fourier shell correlation profile of the 3D density maps of (A) toroid-like particles or (B) square-like particles. FSC curve calculated from two 3D volumes is shown in green or blue and the half-bit curve in red. The global resolution determined based on the half-bit criterion is indicated [88].
(TIF)

**S6 Fig. Comparison between 2D particle projections and 2D reprojections of the 3D density map.** Comparison between square-like particles projections or rod-like particles projections (even columns) and their reprojections from the 3D density map obtained in Relion (odd columns) to validate the 3D reconstruction.
(TIF)

**S7 Fig. Protein sequence alignment and secondary structure prediction of the C-terminal region of coronavirus N proteins.** Multiple sequence alignment was performed using Muscle algorithm (3.8) in EMBL-EBI webserver [89]. The alignment was submitted to Ali2D webserver [90,91] and secondary structure prediction for each sequence was generated using PSIPRED algorithm. The multiple sequence alignment and secondary structure prediction was plotted using 2dSS webserver [92]. The numbers indicate SARS-CoV-2 amino acid position and does not consider the gaps. SARS-CoV-2 domains are annotated on the top of alignments.
(TIF)

**S8 Fig. Analysis of the secondary structure content change along the C-terminal tail dimer simulations.** Secondary structure variation by residue for each monomer along simulation time for each of the five replicates and each monomer. The bar shows the color for each secondary structure. The figure was made using *secstruct* command from CPPTRAJ, a Ambertools20's program, and GNUPLOT.
(TIF)

**S9 Fig. Conformational ensemble analysis of the C-terminal dimer molecular dynamics simulations.** (A) Distribution of radius of gyration of the C-terminal dimeric complex obtained from the entire trajectory. The vertical dashed red line corresponds to the 20 Å cutoff used to select the bound configurations and filter out structures of C-terminal monomers that were not interacting. (B) Principal component analysis obtained from the atomic coordinates of C-terminal bounded configurations. The components PC1 and PC2 were submitted to a clustering analysis with DBSCAN method. The colored points correspond to the six clusters and gray points corresponds to noise. (C) Aligned structures of each ensemble (left) and the representative structure with the lower RMSD in relation to the average structure of the ensemble.
(TIF)

**S10 Fig. Analysis of the inter-monomer interaction energy contributions of the C-terminal tail dimer along the simulations.** Five independent simulations of the two C-terminal tail monomers in which the dimer is spontaneously formed. The plots show the non-bonded contribution (vdw in red line and electrostatic in black line). The plots suggest the mainly non-

bonded contribution is vdw. Minimum distance (in gray line) corresponds to the minimum distance between N protein monomers.
(TIF)

**S11 Fig. Convergence analysis of the flexible fitting process for the N protein dimer. (A)** Cross-correlation coefficient (CCC) between model and the 3D density map and **(B)** RMSD calculations as convergence criteria. Step 1 and Step 2 are MDFF simulations. In Step 1, only the heavy-atom of the RNA and structured domains of the N protein were included in the energy potential of the map, while heavy-atoms of the protein disorder domains were included for Step 2. Minimization 1 is an energy minimization step (10000 steps) with restraints on the protein backbone atom positions, while Minimization 2 is an energy minimization step without any restrain (10000 steps). The CCC and RMSD calculations for Step 1 only include heavy-atoms of RNA and structured domains of the N protein, and for Step 2, Minimization 1, and Minimization 2, heavy-atoms of the RNA and protein.
(TIF)

**S12 Fig. RMSD calculation for the CG simulations of N protein dimer in the presence of different length RNAs (A-H) or without RNA (I).** For each condition, the entire trajectory was aligned by the CTD atoms of the N protein dimer using the first simulations frame as reference and the RMSD was calculated using all atoms of the N protein dimer and the first simulations frame as reference. Each line corresponds to a replicate.
(TIF)

**S13 Fig. RMSD calculation for the CG simulation of the N protein octamer.** The entire trajectory was aligned by all atoms of the N protein octamer using the first frame as reference and the RMSD calculation was performed using the N protein octamer atoms and the first simulation frame as reference.
(TIF)

**S14 Fig. RMSD calculation for the all-atom simulation of the N protein CTD-tail.** The trajectories were aligned by the backbone atoms of the two CTD-tail monomers the RMSD was calculated using the same backbone selection, and the first frame as reference. Each line corresponds to a replicate.
(TIF)

## Acknowledgments

We thank LNNano/CNPEM for the use of electron microscopy facility (TEM-26919, TEM-27882).

## Author Contributions

**Conceptualization:** Marcio Chaim Bajgelman, Rafael Elias Marques, Daniela Barretto Barbosa Trivella, Celso Eduardo Benedetti, Paulo Sergio Lopes-de-Oliveira.

**Data curation:** Helder Veras Ribeiro-Filho, Gabriel Ernesto Jara, Fernanda Aparecida Heleno Batista, Celisa Caldana Costa Tonoli, Adriana Santos Soprano, Antonio Carlos Borges, Alexandre Cassago.

**Formal analysis:** Helder Veras Ribeiro-Filho, Gabriel Ernesto Jara, Fernanda Aparecida Heleno Batista, Gabriel Ravanhani Schleder, Celisa Caldana Costa Tonoli.

**Funding acquisition:** Marcio Chaim Bajgelman, Rafael Elias Marques, Daniela Barretto Barbosa Trivella, Kleber Gomes Franchini, Ana Carolina Migliorini Figueira, Paulo Sergio Lopes-de-Oliveira.

**Investigation:** Helder Veras Ribeiro-Filho, Gabriel Ernesto Jara, Fernanda Aparecida Heleno Batista, Gabriel Ravanhani Schleder, Celisa Caldana Costa Tonoli, Adriana Santos Soprano, Samuel Leite Guimarães, Antonio Carlos Borges, Alexandre Cassago.

**Methodology:** Helder Veras Ribeiro-Filho, Gabriel Ernesto Jara, Ana Carolina Migliorini Figueira, Celso Eduardo Benedetti, Paulo Sergio Lopes-de-Oliveira.

**Project administration:** Daniela Barretto Barbosa Trivella, Kleber Gomes Franchini, Ana Carolina Migliorini Figueira, Celso Eduardo Benedetti, Paulo Sergio Lopes-de-Oliveira.

**Resources:** Marcio Chaim Bajgelman, Rafael Elias Marques, Daniela Barretto Barbosa Trivella, Kleber Gomes Franchini, Ana Carolina Migliorini Figueira, Celso Eduardo Benedetti, Paulo Sergio Lopes-de-Oliveira.

**Software:** Helder Veras Ribeiro-Filho, Gabriel Ernesto Jara, Paulo Sergio Lopes-de-Oliveira.

**Supervision:** Marcio Chaim Bajgelman, Rafael Elias Marques, Daniela Barretto Barbosa Trivella, Kleber Gomes Franchini, Ana Carolina Migliorini Figueira, Celso Eduardo Benedetti, Paulo Sergio Lopes-de-Oliveira.

**Validation:** Daniela Barretto Barbosa Trivella, Ana Carolina Migliorini Figueira, Celso Eduardo Benedetti, Paulo Sergio Lopes-de-Oliveira.

**Visualization:** Helder Veras Ribeiro-Filho, Gabriel Ernesto Jara, Fernanda Aparecida Heleno Batista, Gabriel Ravanhani Schleder, Ana Carolina Migliorini Figueira, Celso Eduardo Benedetti, Paulo Sergio Lopes-de-Oliveira.

**Writing – original draft:** Helder Veras Ribeiro-Filho, Gabriel Ernesto Jara, Ana Carolina Migliorini Figueira, Celso Eduardo Benedetti, Paulo Sergio Lopes-de-Oliveira.

**Writing – review & editing:** Helder Veras Ribeiro-Filho, Gabriel Ernesto Jara, Fernanda Aparecida Heleno Batista, Gabriel Ravanhani Schleder, Celisa Caldana Costa Tonoli, Adriana Santos Soprano, Samuel Leite Guimarães, Antonio Carlos Borges, Alexandre Cassago, Marcio Chaim Bajgelman, Rafael Elias Marques, Daniela Barretto Barbosa Trivella, Kleber Gomes Franchini, Ana Carolina Migliorini Figueira, Celso Eduardo Benedetti, Paulo Sergio Lopes-de-Oliveira.

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
