## [Decision Letter · Decision Letter 0]

25 Feb 2022

Dear Dr. Oliveira,

Thank you very much for submitting your manuscript "Structural dynamics of SARS-CoV-2 nucleocapsid protein induced by RNA binding" for consideration at PLOS Computational Biology. As with all papers reviewed by the journal, your manuscript was reviewed by members of the editorial board and by several independent reviewers. The reviewers appreciated the attention to an important topic. Based on the reviews, we are likely to accept this manuscript for publication, providing that you modify the manuscript according to the review recommendations.

Sincerely,

Alexey Onufriev

Associate Editor

PLOS Computational Biology

Nir Ben-Tal

Deputy Editor

PLOS Computational Biology

[LINK]

Reviewer's Responses to Questions

**Comments to the Authors:**

Reviewer #1: In their manuscript, Ribeiro Filho and colleagues describe a thorough study of the coronavirus nucleocapsid protein structure and dynamics. The authors use electron microscopy to get initial insights into the protein organization, and then use modeling to gain understanding at the molecular level. They show that addition of RNA causes compaction of the nucleoprotein particles, in agreement with experimental data, and characterize the protein-RNA complexes containing either a dimer of the N protein or a tetramer of the N protein dimers. Overall, this is a nice and well-written manuscript that presents important new findings about the N protein, which I think should be published after the following minor and technical issues are resolved.

1) Please provide an estimate of the convergence of the simulations reported in Figures 2C, 4, 6B and 7

2) Papers such as this https://doi.org/10.1038/s41467-021-22785-x suggest that significant part of the RNA inside the virion may be double-stranded. Can you provide some discussion and/or opinion whether the N protein also binds double-stranded RNA and how this could affect the structure of RNPs? Does your model agrees with the findings by Caruso et al? (https://doi.org/10.1016/j.bpj.2021.06.003 and https://doi.org/10.1101/2021.07.21.453232)

3) Have any specific N-RNA interactions been reported? Could they affect the structure of RNPs?

4) Could it also be that the dimers within an octamer interact through RNA?

5) Can you please discuss the possible role of N protein interactions with other viral or cellular proteins (such as nsp3 https://doi.org/10.1126/sciadv.abm4034 or 14-3-3 https://doi.org/10.1016/j.jmb.2021.166875)? Could it be that these proteins are also included in the assembled virion and affect the structure of RNPs?

page 11, sentence "By fitting the atomic model derived from the CG simulations performed with the 60 nt-long RNA into the 27 Å resolution density map (Figure S5A)" refers to incorrect Figure, should be Figure S4B?

page 12, references to the "right inset": perhaps it woult be nicer to clarify "right top inset", "right bottom inset"

page 13, "This finding was crucial because suggested that square and rod-like particles correspond to different orientations": it is also possible that rod-like particles are simply dimers (of dimers) and not tetramers (of dimers) viewed along their plane

page 13, "3D density map for the square-like particles (Figure 5C and Figure S5)" probably should be "Figure 6C and Figure S5"

Reviewer #2: I have three minor questions/concerns:

1) has the procedure for fitting the CG model been tested elsewhere? If so, it should be cited and discussed. If no, it should be tested if possible on another system for accuracy and minimally discussed as a new method.

2) from your work, you should be able to get information on the conformational ensemble. Especially if you combined this work with non-parametric clustering. Is there a reason this wasn't done?

3) see comment under data sharing.

**Have the authors made all data and (if applicable) computational code underlying the findings in their manuscript fully available?**

Reviewer #1: **No: **The simulation data are not yet available without restriction; the electron microscopy data are seemingly not deposited to EMDB (https://www.ebi.ac.uk/emdb/)

Reviewer #2: **No: **the data is claimed to be in a repository; but that repository is locked and I can't get access without revealing my identity as a reviewer. Also, not sure if the repository would contain the "in-house " scripts.

PLOS authors have the option to publish the peer review history of their article (what does this mean?). If published, this will include your full peer review and any attached files.

Reviewer #1: No

Reviewer #2: No

Figure Files:

Data Requirements:

Reproducibility:

References:

---

## [Decision Letter · Decision Letter 1]

19 Apr 2022

Dear Dr. Oliveira,

We are pleased to inform you that your manuscript 'Structural dynamics of SARS-CoV-2 nucleocapsid protein induced by RNA binding' has been provisionally accepted for publication in PLOS Computational Biology.

Best regards,

Alexey Onufriev

Associate Editor

PLOS Computational Biology

Nir Ben-Tal

Deputy Editor

PLOS Computational Biology

Reviewer's Responses to Questions

**Comments to the Authors:**

Reviewer #1: In the revised version, the authors have addressed all of my comments, and I recommend acceptance of the manuscript.

**Have the authors made all data and (if applicable) computational code underlying the findings in their manuscript fully available?**

Reviewer #1: Yes

PLOS authors have the option to publish the peer review history of their article (what does this mean?). If published, this will include your full peer review and any attached files.

Reviewer #1: No

---

## [Editor Report · Acceptance letter]

6 May 2022

PCOMPBIOL-D-22-00057R1 

Structural dynamics of SARS-CoV-2 nucleocapsid protein induced by RNA binding

Dear Dr Oliveira,

I am pleased to inform you that your manuscript has been formally accepted for publication in PLOS Computational Biology. Your manuscript is now with our production department and you will be notified of the publication date in due course.

With kind regards,

Olena Szabo
